# Low-Fidelity Video Encoder Optimization for Temporal Action Localization

**Mengmeng Xu**[1,2*]
mengmeng.xu@kaust.edu.sa

**Juan-Manuel Pérez-Rúa**[1]
PerezRua.JM@gmail.com

**Xiatian Zhu**[1]
xiatian.zhu@samsung.com

**Bernard Ghanem**[2]
bernard.ghanem@kaust.edu.sa

**Brais Martinez**[1]
brais.a@samsung.com

[1] Samsung AI Centre Cambridge, UK [2] King Abdullah University of Science and Technology, Saudi Arabia

## Abstract

Most existing temporal action localization (TAL) methods rely on a transfer learning pipeline, first optimizing a video encoder on a large action classification dataset (*i.e.*, source domain), followed by freezing the encoder and training a TAL head on the action localization dataset (*i.e.*, target domain). This results in a task discrepancy problem for the video encoder – trained for action classification, but used for TAL. Intuitively, joint optimization with both the video encoder and TAL head is an obvious solution to this discrepancy. However, this is not operable for TAL subject to the GPU memory constraints, due to the prohibitive computational cost in processing long untrimmed videos. In this paper, we resolve this challenge by introducing a novel low-fidelity (LoFi) video encoder optimization method. Instead of always using the full training configurations in TAL learning, we propose to reduce the mini-batch composition in terms of temporal, spatial or spatio-temporal resolution so that jointly optimizing the video encoder and TAL head becomes operable under the same memory conditions of a mid-range hardware budget. Crucially, this enables the gradients to flow backwards through the video encoder conditioned on a TAL supervision loss, favourably solving the task discrepancy problem and providing more effective feature representations. Extensive experiments show that the proposed LoFi optimization approach can significantly enhance the performance of existing TAL methods. Encouragingly, even with a lightweight ResNet18 based video encoder in a single RGB stream, our method surpasses two-stream (RGB + optical flow) ResNet50 based alternatives, often by a good margin. Our code is publicly available at https://github.com/saic-fi/lofi_action_localization .

## 1   Introduction

Video analysis has become an increasingly important area of research, encompassing multiple relevant problems such as action recognition [8, 13], temporal action localization [68, 7, 12, 22, 59, 60, 55, 67, 15], video grounding[36, 65, 64, 66, 49, 16, 24], and video question answering [25, 31]. Among those, temporal action localization (TAL) [68, 22] is a fundamental task, as natural videos are not temporally trimmed. Given an untrimmed video, TAL aims to identify the start and end points of all action instances and recognize their category labels simultaneously. A typical TAL model is based on deep convolutional neural networks (CNNs) composed of two modules: a video encoder and a TAL head. The video encoder is often shared across different TAL methods (*e.g.*, G-TAD [60], BC-GNN [3])

---

*Work done during an internship at Samsung AI Centre.

35th Conference on Neural Information Processing Systems (NeurIPS 2021).

by taking a specific off-the-shelf action classification model (*e.g.*, C3D [51], I3D [8], TSM [33]), with the differences residing only in the TAL head. However, instead of short (*e.g.*, 10 seconds) trimmed video clips as in action recognition, the input videos to a TAL model are characterized by much longer temporal duration (*e.g.*, 120 seconds). This causes unique computational challenges that remain unsolved, particularly in model optimization.

In standard optimization of a TAL model, a two-stage transfer learning pipeline is often involved:

1. First, the video encoder is optimized on a large source video classification dataset (*e.g.*, Kinetics [28]) and, optionally, finetunned on the trimmed version of the target dataset under *action classification supervision*;

2. Second, the video encoder is frozen and the TAL head is optimized on the target action localization dataset (*e.g.*, ActivityNet [22], HACS [68]) under *TAL task supervision*.

With this widely-used TAL training pipeline, the video encoder is only optimal for action classification but not for the target TAL task. Specifically, the video encoder is trained so that different short segments within an action sequence are mapped to similar outputs, thus encouraging insensitivity to the temporal boundaries of actions. This is not desirable for a TAL model. We identify this as ***a task discrepancy problem***. Consequently, the final TAL model could suffer from suboptimal performance.

Indeed, jointly optimizing all components of a CNN architecture end-to-end with the target task's supervision is a common practice, *e.g.*, training models for object detection in static images [18, 46, 38]. Unfortunately, this turns out to be non-trivial for TAL. As mentioned above, model training is severely restricted by the large input size of untrimmed videos and subject to the memory constraint of GPUs. This is why the two-stage optimization pipeline as described above becomes the most common and feasible choice in practice for optimizing a TAL model. On the other hand, existing transfer learning methods mostly focus on tackling the data distribution shift problem across different datasets [70, 50], rather than the task shift problem we study here. Regardless, we believe that solving this limitation of the TAL training design bears a great potential for improving model performance.

In this work, we present a simple yet effective *low-fidelity* (LoFi) video encoder optimization method particularly designed for better TAL model training. It is designed to adapt the video encoder from action classification to TAL whilst subject to the same hardware budget. This is achieved by introducing a simple strategy characterized by a new intermediate training stage where both the video encoder and the TAL head are optimized end-to-end using a lower temporal and/or spatial resolution (*i.e.*, low-fidelity) in the mini-batch construction. Compared to the standard training method, our proposed strategy does not increase the GPU memory standard (often a hard constraint for many practitioners). Crucially, with our LoFi training the gradients back-propagate to the video encoder from a temporal action localization loss whilst conditioned on the target TAL head, enabling the learning of a video encoder sensitive to the temporal localization objective.

We make the following **contributions** in this work. **(1)** We investigate the limitations of the standard optimization method for TAL models, and consider that the task discrepancy problem hinders the performance of existing TAL models. Despite it being a significant ingredient, video encoder optimization is largely ignored by existing TAL methods, left without systematic investigation. **(2)** To improve the training of TAL models, we present a novel, simple, and effective *low-fidelity* (LoFi) video encoder optimization method. It is designed specifically to address the task discrepancy problem with the TAL model's video encoder. **(3)** Extensive experiments show that the proposed LoFi optimization method yields new state-of-the-art performance when combined with off-the-shelf TAL models (*e.g.*, G-TAD [60]). Critically, our method achieves superior efficiency/accuracy trade-off with clear inference cost advantage and good generalizability to varying-capacity video encoders.

## 2 Related Work

**Temporal action localization (TAL) models:** TAL models can be grouped by architectural design pattern into two categories, one-stage and two-stage architectures. One-stage methods, either predict temporal action boundaries or generate proposals, and classify them within the same network [3, 9, 21, 40, 34, 58, 60, 62]. The latter type, two-stage models, generate sets of action proposals (*e.g.*, segments) [6, 12, 14, 23, 39] and then an auxiliary head is used for classification of each proposal into an action class [35, 47, 48, 63, 69]. In this work, rather than introducing a novel model design,

we focus on the training of generic TAL models, with a particular aim to improve the video encoder optimization. This is a relatively less investigated aspect in the TAL literature.

**Video encoders in TAL:**  The video encoder is an indispensable part of a TAL model. Main design choices include the base architecture of video encoder and its optimization procedure. With regards to the architecture, the two-stream Temporal Segment Network (TSN) [54] is one of the most common video encoders in existing TAL methods [3, 34, 35, 60]. Concretely, these works use two TSN networks, one with a ResNet50 [20] backbone trained on RGB video frames, and the other with a BN-Inception backbone [26] trained on optical flow. Other alternatives used as a video encoder for TAL include two-stream I3D model [8] (see [19, 63]) and Pseudo-3D [45] (see [40]).

In terms of optimization, a typical paradigm is two-staged: first pre-training the video encoder and then, in a second stage, training the TAL head of the model with the video encoder fixed. This is constrained by the inherent hardware budget derived from having a large per-video input size. In particular, the video encoder is pre-trained using a cross-entropy loss for *action recognition* on a large-scale video classification dataset such as Kinetics [28, 64]. An optional step is to further pre-train it on the foreground segments of the target TAL dataset [34, 60, 11, 44]. This brings a mismatch between training and inference for the video encoder, which we call a task discrepancy problem. More specifically, although trained to distinguish the content of different action classes, the video encoder is less sensitive to action temporal boundaries and thus less effective for the TAL task. In fact, due to their inherent design, CNNs have limited localization capabilities [41], unless they are augmented with specialized localization-specific layers [37]. Additionally, the action classification task focuses only on the foreground content whilst ignoring per-class background segments, including the transition between foreground and background. In this paper we propose a novel low-fidelity video encoder pre-training method to solve this limitation with existing TAL methods.

While current TAL literature mostly relies on pre-training through supervised learning, the rapid advancement of self-supervised learning makes it a promising alternative [2, 4, 42, 43, 56]. Some works have focused on finding effective temporal-related pretext tasks, from frame ordering learnt through triplets of frames [43], to sorting the frames of a sequence [30], distinguishing whether sequences are played forward or backwards [56] or through playback speed-related pretext tasks [4, 61, 53, 27]. These methods exploit video-specific characteristics to force the network to focus on some sort of semantic content within video, inducing representations capturing long-term temporal semantic relations, but force invariance to or ignore the relative positioning of the snippets within the action instances. They are thus not suited for pre-training the video encoder of a TAL model.

Very recent works [1, 59] have exploited some of the aforementioned techniques for better pre-training of action localization models. For example, localization-tailored data augmentation and classification is adopted by [59]. However, these works introduce a large amount of extra video data and additional stream networks, both of which are expensive in terms of memory and computation. In contrast, our method aims to improve TAL modelling directly without the need for learning from extra training video data and using an expensive second network, nor relying on optical flow obtained at high computational cost.

## 3   Method

A TAL model takes as input a long untrimmed video with a varying number of frames. For design convenience, it is typical to represent a varying-length video by decomposing it into a fixed-length sequence of $L$ *snippets*. The definition of a snippet is the same as in action recognition, where first a number of consecutive frames (*e.g.*, 64) is selected and then sub-sampled with stride $r$ (*e.g.*, stride 8 to obtain 8-frame snippets). To represent a snippet, one first applies a video encoder to extract frame-level feature vectors and then averages them to obtain the snippet-level feature representation [6, 12, 17, 35]. The resulting snippet feature sequence is denoted as $X \in \mathbb{R}^{C \times L}$, where $C$ is the feature dimension of each snippet, and $L$ is the number of snippets.

In the training set, each video is associated with its ground truth, consisting of a set of action instance annotations $\Psi$. In particular, each action instance is represented as a segment, each including the start time, the end time, and the action class label. The objective is to train a TAL model that can accurately localize all the target action instances in a given untrimmed video. To that end, the model predicts a

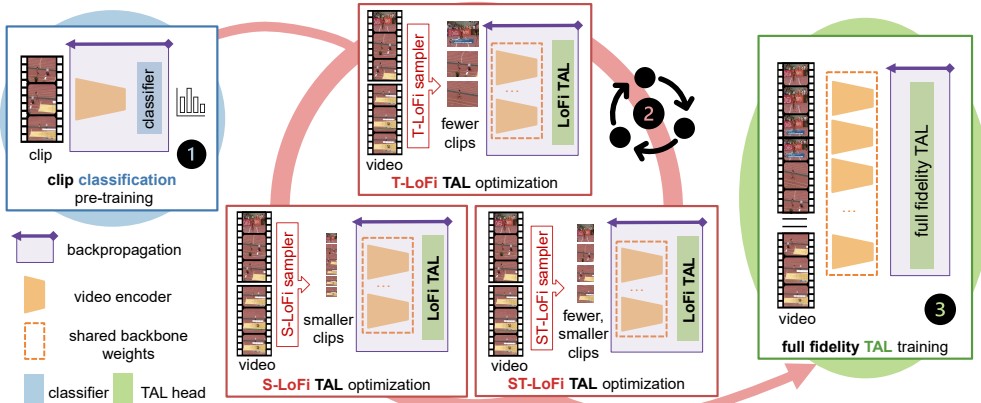

Figure 1: **Schematic overview of the proposed TAL model training procedure.** Three stages are involved during model training: (1) Pre-training the *video encoder* under action classification task's supervision on an auxiliary video dataset (*e.g.*, Kinetics [28]); (2) Low-fidelity (low mini-batch configuration in the spatial, temporal, or spatio-temporal resolution of training videos) optimization of the *video encoder* together with the TAL head under TAL task's supervision on the target dataset; This is the key stage introduced in this paper for resolving the task discrepancy problem without memory overhead increase. (3) Training the *TAL head* in the full fidelity configuration under TAL task's supervision on the target dataset.

varying number of action instances $\Phi$, each comprised of the predicted temporal boundaries, action class, and confidence score.

## 3.1 Model Training Procedure

Our training procedure consists of three stages as depicted in Figure 1. **(1)** First, we pre-train the video encoder by action classification supervision on a large video dataset (*e.g.*, Kinetics [28]). **(2)** Second, we conduct low-fidelity (LoFi) optimization of the video encoder with a *TAL head* on the *target dataset* (Sec. 3.2). This is under TAL supervision with the objective loss function derived from the ground-truth $\Psi$ and model prediction $\Phi$. To this end, we propose to reduce the mini-batch configuration in terms of the spatial and temporal resolution of the input as otherwise end-to-end optimization cannot satisfy the hardware constraints. Crucially, by training on the target task and target dataset, the task discrepancy gap can be reduced. **(3)** Last, we freeze the already end-to-end optimized video encoder and train the TAL head of choice from scratch on the target dataset at full spatial and temporal resolutions. Note that in this setup, we cannot perform end-to-end optimization of the final TAL model, limited by the hardware memory constraint. Next, we will detail the proposed LoFi training.

## 3.2 Low-Fidelity Training Configurations

Formally, we define the *full fidelity* configuration of a mini-batch as

$$\Omega_f = L \times H \times W, \tag{1}$$

where $L$ specifies the temporal resolution, and $H \times W$ refers to the 2D spatial resolution. Under the full fidelity regime and with a certain memory size constraint, only the TAL head can be trained whilst leaving the video encoder frozen. In order to enable the video encoder to be optimized end-to-end together with the TAL head under the TAL task's supervision, we design four low-fidelity configurations for the mini-batch.

**(I) Spatial Low-Fidelity (S-LoFi)** In the first configuration, we lower the spatial resolution of the input videos by a factor of $r_s$ in both spatial dimensions as:

$$\Omega_s = L \times (H/r_s) \times (W/r_s) \tag{2}$$

With smaller spatial feature maps, this could effectively reduce the memory consumption of the video encoder's feature maps which in turn creates space to enable the learning of the video encoder.

**(II) Temporal Low-Fidelity (T-LoFi)**    In the second configuration, we instead consider temporal resolution reduction in the following form:

$$\Omega_t = (L/r_t) \times H \times W, \tag{3}$$

where $r_t > 1$ is the scaling factor. This corresponds to a smaller number of snippets being taken as input by the TAL head, finally outputting less predictions in case the candidates per time location remains the same. This reduces the memory demands of *both* the video encoder and the temporal localization head.

**(III) Spatio-Temporal Low-Fidelity (ST-LoFi)**    In the third configuration, we apply a (typically smaller) reduction in both temporal and spatial resolutions concurrently, formulated as:

$$\Omega_{st} = (L/r_t) \times (H/r_s) \times (W/r_s) \tag{4}$$

In this setup, memory saving can be shared between the temporal and the spatial dimensions.

**(IV) Cyclic Low-Fidelity (C-LoFi)**    Each of the above three LoFi configurations is used in isolation. To further explore their complementary benefits, we propose to apply all of them in a structured fashion. To that end, we take inspiration from the recently proposed multi-grid training strategy [57], originally designed for speeding-up action recognition training. Specifically, we form a sampling grid with the three LoFi configurations and cycle through them repeatedly. We integrate both the short and long cycle strategies [57]. In particular, the **long cycle** changes the values of $r_t$ and $r_s$ at the beginning of an epoch, and cycling through the configurations I to III after $c_l$ epochs. Instead, the **short cycle** also cycles through configurations I to III, but changes $r_t$ and $r_s$ after $c_s$ batches instead.

It is important to note some practical differences with the original multi-grid training. First, contrary to the original multi-grid method, we shift the reduction in input resolution between the temporal and the spatial dimensions so that the *per-video* memory usage remains (approximately) constant. Thus, the batch size also remains constant throughout, which simplifies training. Second, the full resolution setting (*i.e.*, the full fidelity) is not used at any time during the video encoder training.

We compare the above four configurations in Section 4.3. By default, we use the C-LoFi configuration with the long cycle strategy (Long C-LoFi).

### 3.3   An Instantiation of LoFi

**Off-the-shelf TAL model:**    Without loss of generality, in this study we adopt G-TAD [60], a state-of-the-art TAL method, as our temporal localization module. However, any other standard alternative, *e.g.* [34], can be adopted without any additional considerations (see Table A in *supplementary material*). Relying on graph convolutions [29], G-TAD is composed of a stack of GCNeXt blocks to obtain context-aware features. In each GCNeXt block, there are two graph convolution streams to model two types of contextual information. One stream operates with temporal neighbors, and the other adaptively aggregates semantic neighbors in the snippet feature space. At the end of the last GCNeXt block, G-TAD extracts a set of sub-graphs based on pre-defined temporal anchors. With the sub-graph of interest alignment layer, SGAlign, it represents each sub-graph using a feature vector, which is further used as input to multiple fully-connected layers to predict the final action predictions.

To train G-TAD with our LoFi, we select one of the proposed low-fidelity variants and apply the original TAL loss function to optimize both the video encoder and the TAL head. We initialize G-TAD weights as in [60].

**Implementation details:**    We use ResNet-based TSM [33] as the video encoder due to its good accuracy-cost trade-off and reasonable memory requirements compared to 3D-based alternatives. For the full fidelity setting (Eq. (1)), we follow the standard G-TAD protocol and represent each video with $L = 100$ snippets. The full spatial resolution is $H \times W = 224 \times 224$. We keep the other hyper-parameters (*e.g.*, the number of GCNeXt layers) the same as in the default G-TAD configuration. However, the number of anchor proposals can be reduced when $L$ is less. Concretely, we enumerate all the possible combinations of start and end as the anchors, *e.g.*, $\{(t_s, t_e) \mid 0 < t_s < t_e < L;\ t_s, t_e \in \mathcal{N};\ t_e - t_s < L\}$.

For LoFi training, we use an SGD optimizer. The batch size is 16 for all the training methods and input patterns. The weight decay is $10^{-4}$ and we set the momentum to 0, which is standard for fine-tuning [32]. The learning rate is 0.1, and it is decayed by 0.5 after every 5 epochs.

When we train G-TAD using the full fidelity setting, we keep the same training strategy as described in the original paper [60], except that we perform a learning rate search within the set {0.0002, 0.0005, 0.001, 0.002, 0.005}. We follow the rest of the common post-processing steps for TAL as specified in the original paper, including the application of soft-NMS [5] with a threshold of 0.84. We select the top-100 predictions for the final evaluation. We provide our code and script as part of the *supplementary material*.

**Hardware and software settings:** We implemented our method using PyTorch 1.8 with CUDA 10.1. For LoFi training, we use 4 NVIDIA V100 GPUs, each with 32GB memory. Under this setting, the memory constraint is 128GB, which constitutes a mid-range computational budget. In the *supplementary material*, we further test a low-budget setting with a single V100 GPU in Table B.

## 4 Experiments

### 4.1 Experimental setup

**Datasets:** We use ***Kinetics400*** [28] as the auxiliary video classification dataset for initial pre-training of the video encoder. For model performance evaluation, we use two popular temporal action localization benchmarks. (1) ***ActivityNet-v1.3*** [22] contains 20K temporally annotated untrimmed videos with 200 action categories. In the standard evaluation protocol, these videos are divided into the training/validation/testing sets by the ratio of 2:1:1. (2) Human Action Clips and Segments (***HACS-v1.1***) [68] is a recent temporal action localization dataset. It contains 140K complete action segments from 50K videos including 200 action categories (the same ones as ActivityNet-v1.3).

**Evaluation metrics:** We adopt the mean Average Precision (mAP) rate at specified IoU thresholds as the main evaluation metrics. Following the standard evaluation setting, mAP values at a set of IoU thresholds, $\{0.5, 0.75, 0.95\}$ are reported, as well as the average mAP over 10 different IoU thresholds $[0.5 : 0.05 : 0.95]$.

### 4.2 Pre-training Methodology Comparisons

**Setting:** In this set of experiments, we directly compare different methodologies for pre-training the video encoder. To compare with our proposed LoFi method, we first consider the most widely used pre-training method that optimizes the video encoder through the action classification task on an auxiliary dataset (Kinetics400 in our case). We denote this method as *Action Classification Pre-training* (**ACP**). To demonstrate the effect of video encoder pre-training, we also take into account an ImageNet pre-trained video encoder, which uses no video data. We refer to this baseline as *Image Classification Pre-training* (**ICP**). To adapt the video encoder to the target dataset, it can be further trained through an action classification task on a clip version of the target TAL dataset, using each positive segment as a training instance. We denote this variant as **ACP+**.

In this experiment, we train our model using the cyclic low-fidelity (C-LoFi) configuration with the long cycle strategy. Other configurations will be evaluated in Section 4.3.

Table 1: **Comparing TAL results of different video encoder pre-training methods.** ACP: Action Classification Pre-training, ICP: Image Classification Pre-training, ACP+: further fine-tuning ACP on the target dataset using a classification task using positive action segments.

| Metric | 0.5 | 0.75 | 0.95 | Average | 0.5 | 0.75 | 0.95 | Average |
|---|---|---|---|---|---|---|---|---|
| Dataset | | ActivityNet-v1.3 | | | | HACS-v1.1 | | |
| ACP | 49.64 | 34.16 | 7.68 | 33.59 | 35.68 | 22.79 | 6.51 | 23.00 |
| ICP | 48.45 | 32.40 | 6.89 | 32.16 | 31.74 | 19.64 | 5.67 | 20.18 |
| ACP+ | 49.87 | 34.58 | 7.85 | 33.84 | 36.31 | 22.96 | 6.62 | 23.31 |
| **LoFi (ours)** | **50.68** | **35.16** | **8.16** | **34.49** | **37.47** | **24.36** | **7.08** | **24.62** |

**Results:** The results of the different video encoder pre-training methods are reported in Table 1. We make the following observations. **(1)** Without pre-training on a related auxiliary dataset, the model

performance could be significantly degraded (see row 1 *v.s.* row 2). This indicates the significance of the video encoder and its pre-training on large, relevant video data. **(2)** With action classification based pre-training on the target video data, the performance indeed improves to some degree (see row 1 *v.s.* row 3). This means that addressing the data distribution shift between the auxiliary dataset and the target dataset is essential. **(3)** Importantly, the biggest performance gains come from the proposed LoFi method, which instead solves the task discrepancy issue by optimizing the video encoder with the TAL task (see the last row). This indicates that although auxiliary video data is similar to the target data in distribution, the task-level differences would still pose obstacles harming the model performance. This confirms the motivation and hypothesis of this study. Once this obstacle is properly tackled with our low-fidelity pre-training, more significant performance gains can then be rewarded. Overall, this verifies the efficacy of the proposed method in pre-training the video encoder.

## 4.3 LoFi Configuration Comparisons

**Setting:** We investigate different LoFi configurations. We use the setting as: $r_t = 4$ for T-LoFi, $r_s = 2$ for S-LoFi (note that this is applied in both spatial dimensions, thus being comparable to T-LoFi), $r_s = \sqrt{2}, r_t = 2$ for ST-LoFi, and $c_s = 16$, $c_l = 1$ for C-LoFi's short and long cycle strategies, respectively. We include the standard pre-training strategy (*i.e.*, ACP) to provide a baseline and facilitate comparisons.

Table 2: **Comparing different low-fidelity configurations**. Dataset: ActivityNet-v1.3 and HACS-v1.1. ACP: Action Classification Pre-training. TR: Temporal Resolution; SR: Spatial Resolution.

| Configuration | TR | SR | ActivityNet-v1.3 | | | | HACS-v1.1 | | | |
| --- | --- | --- | --- | --- | --- | --- | --- | --- | --- | --- |
| | | | 0.5 | 0.75 | 0.95 | Avg | 0.5 | 0.75 | 0.95 | Avg |
| ACP | - | 224×224 | 49.64 | 34.16 | 7.68 | 33.59 | 35.68 | 22.79 | 6.51 | 23.00 |
| S-LoFi | 100 | 112×112 | 50.47 | 34.71 | 7.57 | 34.12 | 37.16 | 24.23 | 6.84 | 24.25 |
| T-LoFi | 25 | 224×224 | 50.28 | 35.21 | 8.09 | 34.32 | 37.30 | 24.18 | 7.07 | 24.36 |
| ST-LoFi | 50 | 158×158 | 50.36 | 34.79 | 7.59 | 34.12 | 37.13 | 24.27 | 7.08 | 24.36 |
| Short C-LoFi | Batch-level Cycle | | 50.57 | 35.12 | 8.14 | 34.38 | 37.63 | **24.45** | 6.95 | 24.60 |
| Long C-LoFi | Epoch-level Cycle | | **50.68** | **35.16** | **8.16** | **34.49** | **37.78** | 24.40 | **7.29** | **24.64** |

**Results:** The results for ActivityNet-v1.3 and HACS-v1.1 are shown in Table 2. We provide the following observations. **(1)** Each of our proposed LoFi configurations can improve the video encoder, thus suggesting that spatial and temporal dimensions are both good selections for low-fidelity manipulation. **(2)** Integrating our LoFi configurations into a more advanced cyclic training strategy produces some further, although moderate, improvement. In particular, the long cycle leads to the best overall performance for both datasets.

Table 3: **Comparing TAL results with state-of-the-art methods on ActivityNet-v1.3 validation set**. "*" indicates RGB-only Kinetics400 pre-trained TSM video encoder without fine-tuning. O.F.: Optical Flow. R18/50: ResNet-18/50.

| Method | O.F. | Arch. | #Par | 0.5 | 0.75 | 0.95 | Average |
| --- | --- | --- | --- | --- | --- | --- | --- |
| SCC [21] | ✗ | C3D | 79M | 40.00 | 17.90 | 4.70 | 21.70 |
| CDC [47] | ✗ | C3D | 79M | 45.30 | 26.00 | 0.20 | 23.80 |
| R-C3D [58] | ✗ | C3D | 79M | 26.80 | - | - | - |
| BSN [35] | ✓ | R50 | 23M | 46.45 | 29.96 | 8.02 | 30.03 |
| P-GCN [63] | ✓ | I3D | 25M | 48.26 | 33.16 | 3.27 | 31.11 |
| BMN [34] | ✓ | R50 | 23M | 50.07 | 34.78 | 8.29 | 33.85 |
| BC-GNN [3] | ✓ | R50 | 23M | 50.56 | 34.75 | **9.37** | 34.26 |
| G-TAD [60] | ✓ | R50 | 23M | 50.36 | 34.60 | 9.02 | 34.09 |
| G-TAD baseline* | ✗ | R18 | 12M | 49.64 | 34.16 | 7.68 | 33.59 |
| G-TAD+ **LoFi** | ✗ | R18 | 12M | **50.68** | **35.16** | 8.16 | **34.49** |

Table 4: **Comparing TAL results with state-of-the-art methods on HACS-v1.1 validation set**. "*" indicates RGB-only Kinetics400 pre-trained TSM video encoder without fine-tuning. O.F.: Optical Flow. R18/50: ResNet-18/50. 2S.: 2-stream.

| Method | O.F. | Arch | #Par | 0.5 | 0.75 | 0.95 | Average |
|--------|------|------|------|------|------|------|---------|
| SSN [69] | ✓ | 2S | 12M | 28.82 | 18.80 | 5.32 | 18.97 |
| G-TAD baseline* | ✗ | R18 | 12M | 35.68 | 22.79 | 6.51 | 23.00 |
| G-TAD + **LoFi** | ✗ | R18 | 12M | **37.78** | **24.40** | **7.29** | **24.64** |

## 4.4 Comparison with State-of-the-Art

**Setting:** Following video encoder pre-training evaluation, we further conduct a system-level performance comparison with previous state-of-the-art methods. Whilst ResNet50-based (R50) methods are a common backbone choice for the video encoder, we still use a more lightweight ResNet18 (R18) for our method due to its higher efficiency in computation and memory. Furthermore, compared to the popular architecture design that uses two streams (one for RGB and one for optical flow), our method only uses RGB frames, avoiding the excessive costs incurred from computing optical flow and running a second forward pass.

**Results:** The results of our method are compared with existing alternatives in Table 3 for ActivityNet-v1.3 and Table 4 for HACS-v1.1. It is evident that our method can achieve the best performance among all the competitors, despite the single stream input modality and a much lighter video encoder backbone. This clearly demonstrates that tackling the pre-training of the video encoder is of particular importance for TAL, and that existing efforts towards improving the TAL head model have neglected it as a key model component. On ActivityNet-v1.3, it is encouraging to see that with a stronger pre-trained video encoder using our method and a shallower architecture, optical flow can be favourably eliminated without performance sacrifice (actually even enjoying better performance). This result is substantial, since the study of means to get rid of optical flow for more efficient action analysis is itself an important research problem [10].

## 4.5 Using Different Video Encoders

**Setting:** Our LoFi method can be used in combination with different video encoders, as long as the backbone is end-to-end trainable. We compare the performance gains of our default encoder, *i.e.* ResNet18-based TSM, with a 3D-based alternative, *i.e.* an 18-layer R(2+1)D encoder [52], on ActivityNet-v1.3. Note that the default spatial resolution for R(2+1)D, as defined by the authors, is $112 \times 112$ pixels. We thus use T-LoFi and maintain the default spatial resolution for both networks.

Table 5: **Ablation on different video encoders.** The performance improvement from using our LoFi pre-training does not depend on the video encoder's backbone. Dataset: ActivityNet-v1.3. TAL head: G-TAD. R18: ResNet18.

| Method | TSM-R18 backbone | | | | R(2+1)D-R18 backbone | | | |
|--------|------|------|------|------|------|------|------|------|
| | 0.5 | 0.75 | 0.95 | Avg | 0.5 | 0.75 | 0.95 | Avg |
| G-TAD + ACP | 49.64 | 34.16 | 7.68 | 33.59 | 49.65 | 34.11 | **8.66** | 33.55 |
| G-TAD + T-LoFi | **50.28** | **35.21** | **8.09** | **34.32** | **49.84** | **34.73** | 8.64 | **34.21** |
| *Gain* | +0.64 | +1.05 | +0.41 | +0.73 | +0.19 | +0.62 | -0.02 | +0.66 |

**Results:** The results are summarized in Table 5. We observe that with either TSM-R18 or R(2+1)D as video encoder, our method can similarly and consistently improve the performance of the state-of-the-art G-TAD method. This verifies that our LoFi method is generally effective and useful in training TAL models.

### 4.6 Scaling up to ResNet-34 and ResNet-50

**Setting:** To evaluate the generalizability of our LoFi, we further evaluate two deeper video encoders: ResNet34/50-based TSM. We use the long cyclic low-fidelity configuration. Limited by the GPU memory size, we reduce the batch size accordingly whilst keeping all other hyper-parameters and the training protocol unchanged. This experiment is conducted on ActivityNet-v1.3 in comparison with the standard action classification pre-training (ACP) method as baseline.

Table 6: **Ablation on different depths of TSM video encoders.** The performance improvement from using our LoFi pre-training does not depend on the video encoder's backbone. Dataset: ActivityNet-v1.3. TAL head: G-TAD. R18: ResNet18; R34: ResNet34; R50: ResNet50.

| Encoder | Method | 0.5 | 0.75 | 0.95 | Avg. (*Gain*) |
|---|---|---|---|---|---|
| TSM-R18 | ACP | 49.64 | 34.16 | 7.68 | 33.59 |
| TSM-R18 | **Long C-LoFi** | **50.68** | **35.16** | **8.16** | **34.49** (+0.90) |
| TSM-R34 | ACP | 50.16 | 34.35 | 8.31 | 33.90 |
| TSM-R34 | **Long C-LoFi** | **50.79** | **35.39** | **8.38** | **34.74** (+0.84) |
| TSM-R50 | ACP | 50.32 | 35.07 | 8.02 | 34.26 |
| TSM-R50 | **Long C-LoFi** | **50.91** | **35.86** | **8.79** | **34.96** (+0.70) |

**Results:** Table 6 shows that whilst the gains with deeper encoders are slightly reduced, LoFi can still consistently improve the results. This suggests good generalizability properties across varying-capacity networks.

### 4.7 Comparison to Self-Supervised Learning

**Setting:** We compare our method to two representative video self-supervised learning methods: Arrow of Time [56] and SpeedNet [4]. For both competitors, TSM-R18 is used as video encoder, and pre-trained on the Kinetics400 dataset. We use ActivityNet-v1.3 dataset for this experiment.

**Results:** From Table 7, we see that both SSL methods without action classification based supervision are clearly inferior for the TAL task (see row 2 and 3). When combined with the standard action classification pre-training (ACP), only slight performance gains can be achieved despite doubling the computational cost in video encoding (see row 4 and 5). This suggests marginal complementary benefit. Overall, this result indicates that the proposed LoFi optimization is superior over recent SSL alternatives in pre-training TAL's video encoder.

Table 7: **Comparison to self-supervised learning methods.** We use TSM-R18 as video encoder. Dataset: ActivityNet.

| Method | 0.5 | 0.75 | 0.95 | Average |
|---|---|---|---|---|
| ACP | 49.64 | 34.16 | 7.68 | 33.59 |
| Arrow [56] | 44.14 | 28.87 | 5.90 | 28.82 |
| Speed [4] | 44.50 | 29.52 | 6.14 | 29.39 |
| ACP+Arrow | 49.79 | 34.48 | 7.70 | 33.72 |
| ACP+Speed | 49.84 | 34.11 | 7.50 | 33.75 |
| **LoFi** | **50.68** | **35.16** | **8.16** | **34.49** |

## 5 Conclusion

In this work we have presented a simple and effective low-fidelity (LoFi) video encoder optimization method for achieving more effective TAL models. This is motivated by an observation that in existing TAL methods, the video encoder is merely pre-trained by action classification supervision on short video clips, lacking desired optimization *w.r.t.* the temporal localization supervision on the target dataset. Indeed, joint optimization itself is not novel. However, this is non-trivial to conduct for training a TAL model, due to large per-video size that would easily overwhelm the GPU memory, rendering it infeasible in practice. To overcome this obstacle, we propose to reduce the mini-batch construction configurations in the temporal and/or spatial dimensions of training videos so that end-to-end optimization becomes operable under the same memory condition. Extensive experiments demonstrate that our method can clearly improve the performance of existing off-the-shelf TAL models, yielding new state-of-the-art performance even with only RGB modality as input and a more lightweight backbone based single-stream video encoder on two representative TAL benchmarks.

# 6 Ethical considerations and broader impact

Any inherent bias present in the training data is likely to be captured by the learning algorithm given its data-driven nature. Deploying of the model into real-world scenarios should thus take that aspect into account. Furthermore, the method discussed in this paper focuses on a fundamental problem and as such its potential applications are hard to predict. However, to the best of our knowledge, there are currently no applications of this technology that raise ethical issues. Finally, biases based on gender, race or sexuality are unlikely, although not impossible.

# 7 Acknowledgements

This work was supported by the King Abdullah University of Science and Technology (KAUST) Office of Sponsored Research through the Visual Computing Center (VCC) funding.

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
