# Low-Fidelity Video Encoder Optimization for Temporal Action Localization
## *(Supplementary Material)*

**Mengmeng Xu**[1,2*]
mengmeng.xu@kaust.edu.sa

**Juan-Manuel Pérez-Rúa**[1]
PerezRua.JM@gmail.com

**Xiatian Zhu**[1]
xiatian.zhu@samsung.com

**Bernard Ghanem**[2]
bernard.ghanem@kaust.edu.sa

**Brais Martinez**[1]
brais.a@samsung.com

[1] Samsung AI Centre Cambridge, UK [2] King Abdullah University of Science and Technology, Saudi Arabia

## 1   Additional Experiments

### 1.1   LoFi optimization works with other TAL models

Our LoFi optimization is a model-agnostic optimization method. In the main paper, we have evaluated LoFi using G-TAD [3] as the TAL head. To evaluate its generality, we further test Boundary Matching Network (BMN) [2] which generates temporal action proposals for an existing action classifier [4] to predict the final results.

**Settings:**   We compare our T-LoFi with the Action Classification Pre-training (ACP) baseline that optimizes the video encoder through the action classification task on an auxiliary dataset (Kinetics400 [1] in our case). We use the same hyper-parameter setting as in G-TAD case, with ResNet-18 as the video encoder backbone. We use publicly available BMN code [2]. As the performance metrics, we follow the standard AR@$k$ (the average recall of the top-$k$ predictions) and AUC (the area under the recall curve).

Table A:   **Comparing TAL results of BMN on ActivityNet-1.3 validation set when using two different video encoder optimization methods (ACP vs. LoFi).**   ACP: Action Classification Pre-training.

| Metric | AR@1 | AR@5 | AR@10 | AR@100 | AUC |
|---|---|---|---|---|---|
| ACP | 33.29 | 48.90 | 56.20 | 74.88 | 66.81 |
| **LoFi (ours)** | **33.71** | **49.41** | **56.81** | **75.58** | **67.49** |
| *gain* | +0.32 | +0.51 | +0.61 | +0.70 | +0.58 |

**Results:**   The results are reported in Table A. We observe that under all the evaluation metrics, our LoFi method consistently improves BMN's performance compared to ACP. Together with the performance gain for G-TAD, this verifies that LoFi is generally effective in improving TAL models.

---

[*]Work done during an internship at Samsung AI Centre.
[2]https://github.com/JJBOY/BMN-Boundary-Matching-Network

35th Conference on Neural Information Processing Systems (NeurIPS 2021).

## 1.2 Performance-Hardware Budget Trade-off

In the main paper, we have conducted the experiments with a fixed computational budget of 4 V100 GPUs. To test our LoFi in varied hardware conditions, we further compare the model performance trade-off under two different budget cases.

**Setting:** For a *low-budget* case, we define a configuration so that the whole training procedure can fit in a single V100 GPU (32GB)[3]. In this setting, the temporal resolution needs to be lowered to 25 snippets and the spatial resolution to $112 \times 112$ pixels. For a *middle-budget* case, we consider 4 V100 GPUs (128G). Under this setting, we further introduce a novel configuration (termed as T-LoFi): using a lower spatial resolution ($112^2$) in exchange for a larger video backbone, ResNet-50.

Table B: **Trade-off analysis between performance and budget (GPU memory) on ActivityNet-1.3 validation set**. TR: Temporal Resolution. SR: Spatial Resolution. R18/50: ResNet-18/50.

| TR/SR | Arch. | GPU | 0.5 | 0.75 | 0.95 | Average |
|---|---|---|---|---|---|---|
| ACP | R18 | – | 49.64 | 34.16 | 7.68 | 33.59 |
| $25/112^2$ | R18 | 32G | 50.01 | 34.46 | **8.38** | 33.99 |
| $25/224^2$ | R18 | 128G | **50.28** | 35.21 | 8.09 | **34.32** |
| $25/112^2$ | R50 | 128G | 50.07 | **35.31** | 8.03 | 34.23 |

**Results:** The performances of our method under different budget settings are compared in Table B. We have these observations: (1) It is seen that our method can still outperform the standard action classification pre-training (ACP) baseline under the low computational budget setting (see the first two rows). (2) For the middle-budget case, it is found that lowering the spatial resolution for using a deeper video encoder (ResNet-50) leads to a slightly worse trade-off in Average-mAP; Besides, using a ResNet18-based video encoder offers a clear efficiency advantage at inference.

## 1.3 Effectiveness of Joint Optimization

While our proposed method effectively closes the domain and task gaps for TAL, it still has to sacrifice input spatial and/or temporal resolution. Thus, although arguably less damaging in terms of the final performance, there still exists a gap between train-time and test-time settings (low-fidelity *v.s.* full-fidelity). In this section, we set both train-time and test-time to low-fidelity settings to evaluate the exact benefit from joint optimization of video encoder and TAL head.

**Setting:** We train G-TAD using 25 snippets (*i.e.*, T-LoFi, $r_t = 4$) in the following two settings: 1) using a video encoder pre-trained on Kinetics400 (*i.e.*, ACP baseline) and 2) using end-to-end training.

Table C: **Comparing the video encoders pre-trained on Kinetics400 and end-to-end training.** We use the LoFi setting (25 snippets, $224^2$ res.)

| Method | Dataset | 0.5 | 0.75 | 0.95 | Avg. (Gain) |
|---|---|---|---|---|---|
| pre-trained | ActivityNet-v1.3 | 45.75 | 32.05 | 4.80 | 31.02 |
| end-to-end | ActivityNet-v1.3 | **47.52** | **33.30** | **5.31** | **32.21** (+1.19) |
| pre-trained | HACS-1.1 | 29.28 | 17.95 | 4.05 | 18.49 |
| end-to-end | HACS-1.1 | **31.08** | **19.74** | **4.29** | **19.94** (+1.45) |

**Results:** The resulting performances are compared for both ActivityNet-v1.3 and HACS-1.1 in Table C. We can see that, while the absolute performance is significantly lower due to the lack of temporal resolution, the accuracy improves significantly on both datasets, namely 1.19 and 1.45 in

---

[3]Admittedly, a V100 GPU is still a high-end GPU, having 32GB of memory. The current on-demand hourly rate at AWS is 3.06USD.

terms of Average-mAP. We also note that higher gains are achieved on HACS-1.1 than on ActivityNet-v1.3. We hypothesize that this is due to the more considerable amount of training data available on HACS-1.1, resulting in more benefits from our LoFi in video encoder pre-training.

## 1.4 Training Stability

In this section, we show our model robustness via the standard derivation of different evaluation metrics. We emphasize that the primary evaluation metric of TAL model is the average of mAP under the ten difference IoU thresholds, and mAP is the mean of Average-Precision over $k = 200$ action classes. Thus, average mAP has a naturally robust property and, more importantly, *a relatively small gain is still significant*.

**Setting:** With G-TAD as the TAL model, we experiment each optimization method (ACP, LoFi) for 10 trials to test their stability. We report the average result with the standard derivation on ActivityNet-1.3 dataset. We use ResNet18 as the video encoder backbone.

Table D: **Comparing TAL results of different video encoder pre-training methods.** ACP: Action Classification Pre-training,

| Metric | 0.5 | 0.75 | 0.95 | Average |
|---|---|---|---|---|
| ACP | $49.64 \pm 0.09$ | $34.16 \pm 0.05$ | $7.68 \pm 0.17$ | $33.59 \pm 0.04$ |
| **LoFi (ours)** | $50.68 \pm 0.12$ | $35.16 \pm 0.07$ | $8.16 \pm 0.16$ | $34.49 \pm 0.03$ |

**Results:** The results in Table D show that both methods are stable with some slight advantage of our LoFi in Average-mAP.

## 2 Further Discussion

### 2.1 Our method's limitations

A main limitation imposed by our proposed LoFi optimization method is that an extra training stage is introduced, which increases the model training complexity. However, due to the nature of low mini-batch configurations, the computational cost is still tolerable.

### 2.2 Social Impact

The research presented in the paper has a potential to positively contribute to a number of practical applications where understanding human's actions and events in video is critical, for example, pedestrian safety in automotive settings, patient monitoring in hospitals and elderly care homes. However, there is also a risk for the technology to be used for nefarious purposes, for example, in the area of unauthorized and immoral surveillance particularly by autocratic regimes. For partial mitigation, we commit to not authorize our technology to be used by any government bodies with such predictable risks.