# OpenReview forum: "Low-Fidelity Video Encoder Optimization for Temporal Action Localization"
_NeurIPS.cc/2021/Conference — NeurIPS 2021 Poster_

### Official Review · Reviewer_mQTb · 2021-07-15

**Rating:** 6
**Confidence:** 3

**Summary:**

Updating after the rebuttal:
The rebuttal well addressed some of my concerns. I would raise the rating to 6: Marginally above the acceptance threshold

---

The paper presents a low-fidelity video encoder optimization approach to relieve the large memory constraints in the temporal action localization problem. The proposed approach is easy to implement and reasonable results have been obtained on ActivityNet and HACS datasets. However, the proposed approach is more like an engineering solution. The technical novelty of the proposed approach is limited.

**Limitations And Societal Impact:**

## Contribution is limited
The paper is mainly based on a new training pipeline with three stages. To releive the high memory cost issue, it proposes to use low-fidelity setting in order to train both video encoder and temporal action localization head. It is more like an engeering solution for this problem. The techinical contribution is not significant.

## Experiments
How about the training time for the proposed approach? With the similar training, how about the performance of baseline.
In Table 6, it presents the results based on different backbone like Resnet 18/34/50. How about the detail setup for this experiment? For example, Res50 should be much larger than Res18, the memory cost should be larger.

In Table 2, the experiments provide several setting for the low-fidelity configurations. Is it possible to provide more results on different setups.

Based on the setup,  4 V100 GPUs have been utilized. How about the performance with more GPUs.


**Main Review:**

## Originality
The originality is below the threshold. The main contribution of the paper is a new training pipeline which is based on the low-fidelity setting to enable joint ttraining of action classification and localization.  It is not sufficient to support a NIPS-level paper.

## Quality
The proposed approach is technical sound. Based on the details in the paper, it should not be difficult to reproduce the results. Also, the proposed pipeline obtain good results on two benchmark datasets like ActivityNet and HACS. Ablation studies are sufficient to support the proposed training pipeline.

## Clarity
The paper is well presented and sufficient details have been provided in the paper.

## Significance
The proposed pipeline should be useful for as a module for other temporal localization algorithms.


**Time Spent Reviewing:**

3

---

> ### Author Response · Authors · 2021-08-10
> **Response to Reviewer mQTb**
>
> We thank the reviewer for the detailed review as well as the suggestions for improvement. Our response to the reviewer’s comments is as below:
>
> ---
>
> Q1: _The main contribution of the paper is more like an engineering solution for this problem._
>
> R1: We have extensive experiments showing that our solution is effective, clearly surpassing more complex solutions. In our opinion, that the proposed solution is also intuitive and simple should be seen as a plus, not a minus. Furthermore, a multitude of papers argue that end-to-end training is not possible for TAL. Challenging this widespread belief is, in our opinion, a contribution of its own. Finally, we would like to stress again the contributions listed in L67-76 in our main paper.
>
> ---
>
> Q2: The proposed pipeline should be useful for as a module for other temporal localization algorithms.
>
> R2: We totally agree -- Our LoFi is agnostic to the specific TAL model employed. We have verified this by using our LoFi training in combination with G-TAD (main experiments), BMN (supplementary material, experiment referred to in L188 of the main paper), and we now have run an experiment comparing BSN and BSN+LoFi. Note that the experiments shown below use RGB features only, not RGB+Flow.
>
> | Method      | Backbone | AR@1 | AR@5 | AR@10 | AR@100 | AUC |
> | ----------- | ----------- | ----------- | ----------- | ----------- | ----------- | ----------- |
> | BSN      | TSM-RN50 | 32.75 | 47.51 | 54.01 | 74.61 | 66.19 |
> | BSN + LoFi   | TSM-RN50        | 32.86 | 48.00 | 55.14 | 75.08 | 66.73 (+0.54) |
>
> ---
>
> Q3: _the training time for the proposed approach, detail setup for experiments In Table 6._
>
> R3: Thanks. We will add more experimental details and a table for training time and memory usage. We train over 20 epochs, taking 6, 9 and 15 hrs of training for ResNet 18, 34 and 50 respectively when using the default hardware configuration described in the main paper.
>
> | Backbone      | Memory/video | Time/epoch |
> | ----------- | ----------- | ----------- |
> | ResNet18 | 8.9GB | 1075 sec |
> | ResNet34 | 11.5GB | 1624 sec |
> | ResNet50 | 29.0GB | 2694 sec |
>
> ---
>
> Q4: _Is it possible to provide more results on different setups In Table 2?_
>
> R4:  In our submission, we tested the feasible and representative configurations we could think of, but we’re very open to testing new configurations if there are specific suggestions.
>
> ---
>
> Q5: _the performance with more GPUs._
>
> R5: This is a good suggestion. We would like to note however that increasing the number of GPUs will require significant changes to the code and computer parallelism, which is a major endeavor for the rebuttal period. We will run this experiment in the near future.

---

### Official Review · Reviewer_krQv · 2021-07-15

**Rating:** 8
**Confidence:** 5

**Summary:**

Models for temporal action localization (TAL) in long, untrimmed video are typically trained in two stages, a video encoder pretrained on a auxiliary video clip classification dataset and then frozen, followed by a TAL head. End-to-end training of the encoder and TAL head is infeasible because the encoder activations for an entire, long video will not fit in GPU RAM. This paper proposes a solution: use the ideas of mulgrid training [51] to do end-to-end training of a lower resolution video encoder and TAL head. They explore strategies for cycling through regimes of lower spatial, temporal, and spatio-temporal resolution training, and demonstrate performance gains on the ActivityNet and HACS datasets, using a number of backbone networks and TAL heads.


**Limitations And Societal Impact:**

Yes, mostly addressed. See discussion of scalability and surveillance in the "Main Review"

**Main Review:**

[Originality] This paper confronts a limitation--the inability to train the video encoder e2e on the TAL task--that is simply accepted in hundreds (?) of TAL papers, and proposes a workable solution. The multigrid methods lying at the heart of this were proposed in another paper about video action classification [51], but here they're applied in a very novel way to address a long-standing problem.

[Clarity] The paper is very well written, and clear. I enjoyed reading it. However it is missing a clear explanation as to why training on lower spatial or temporal resolution inputs would be expected to work - both that the models would be compatible, and that the features learned when r_s = 2 would even make sense for r_1 = 1, for example. I did find this in the "Multigrid Method" paper [51]:

>_"Multigrid training is possible because video models are compatible with input data of variable space and time dimensions due to weight sharing operations (e.g., convolutions). In addition, CNNs are effective at learning patterns at multiple scales, e.g., as observed when training with data augmentation [18, 26, 38]. We observe similar multi-scale robustness and generalization with multigrid training"_

This paper really needs to include a similar argument.

Some additional minor points:
- The "Further Discussion" points in the appends should be in the main paper
- Line 164 confused me "finally outputting less predictions in case the candidates per time location remains the same"
- Line 179 Typo: multi-grid

[Quality] This is a solid, well-supported paper. I think there is a limitation to this approach that is not fully-addressed: in the case of high-parameter SOTA video feature encoder, or when input videos are long, the approach might not work at all because end-to-end training would require such drastically large values of r_s and r_t. I noticed that the authors focus on a very cheap model ResNet18, which seemed like a dodge to the limitation, but felt better when I saw the R(2+1)D 3DCNN experiments. Still, it might be nice to confront the problem directly, do the math about what is feasible, and say whether I the approach will work with SlowFast, a Non-local 3D ResNet-101s, etc.

[Significance] I think this paper has a high degree of novelty, by addressing a long-standing problem in a unique way. The performance numbers are good, not amazing, but certainly enough to demonstrate that this new direction has promise and warrants more attention.


**Time Spent Reviewing:**

4

---

> ### Author Response · Authors · 2021-08-10
> **Response to Reviewer krQv**
>
> We thank the reviewer for the positive and detailed review as well as the suggestions for improvement. Our response to the reviewer’s comments is as below:
>
> ---
>
> Q1: _That is simply accepted in hundreds (?) of TAL papers, and proposes a workable solution._
>
> R1: Thanks for pointing this out. Questioning this statement was indeed our original motivation for this study.
>
> ---
>
> Q2: _why training on lower spatial or temporal resolution inputs would be expected to work._
>
> R2:  Thanks for this important comment. We agree that a similar argument as [51] also applies to our work since both are based on CNN with hierarchical representation structures. We will add this discussion to the revised manuscript. As a small note, our training never uses the full temporal and spatial resolution simultaneously due to hardware memory constraint. Alternating between full spatial resolution and full temporal resolution is however enough to learn these multi-scale (including full-scale) patterns adequately - though this is just empirically verified.
>
> ---
>
> Q3: _The "Further Discussion" points in the appends should be in the main paper; Line 179 Typo: multi-grid_
>
> R3: Thanks. We will fix this.
>
> ---
>
> Q4: _Explanation of Line 164 "finally outputting less predictions in case the candidates per time location remains the same"_
>
> R4: Sorry for the confusion, we will clarify on the revised manuscript. This refers to an implementation detail of G-TAD: the number of candidates is quadratically proportional with the number of snippets. For example, G-TAD yields 300 candidates when using 25 snippets per video and 4950 candidates when using 100 snippets.
>
> ---
>
> Q5: _In the case of high-parameter SOTA video feature encoder, or when input videos are long, the approach might not work at all because end-to-end training would require such drastically large values of r_s and r_t._
>
> R5: This is indeed a valid concern. We believe processing very long videos at a time is in fact a general problem for most TAL methods. An intuitive strategy is to chunk a long video into shorter ones, process each of the shorter videos, and finally merge the per-short-video results [A], or alternatively, to train by sampling more manageable segments of the long video. As such, our LoFi would still apply. But overall we agree that this is a limitation of our method in its current form.
>
> [A] EPIC-KITCHENS-Challenges-2021-Report, Dima Damen, Adriano Fragomeni, Jonathan Munro, Toby Perrett, Daniel Whettam, Michael Wray, https://epic-kitchens.github.io/Reports/EPIC-KITCHENS-Challenges-2021-Report.pdf
>
> ---
>
> Q6: _The authors focus on a very cheap model ResNet18._
>
> R6: We chose ResNet18 for most of our experiments and ablations due to its computational efficiency - working with video is very resource-intensive. However, once we found the best-performing configuration, we tested it with ResNet34 and ResNet50 backbones. These results are included in Table 6 and show consistent improvements across all different network sizes. This is in addition to the R(2+1)D experiments mentioned by the reviewer.
>
> ---
>
> Q7: _Do the math about what large video encoder is feasible._
>
> R7: We have now measured the memory usage of different backbones in the following table. Here OOM means  Out Of Memory, e.g., more than 32 GB in our case.
>
> | Backbone      | With Nonlocal | Memory per video |
> | ----------- | ----------- | ----------- |
> | TSM-RN18      | No       | 8.9 GB |
> | TSM-RN34      | No       | 11.5 GB |
> | TSM-RN34      | Yes       | 16.0 GB |
> | TSM-RN50      | No       | 29.0 GB |
> | TSM-RN50      | Yes       | OOM, ~33 GB |
> | TSM-RN101      | No       | OOM |
> | SlowFast-RN50, 4x16      | No       | 26.1 GB |

---

### Official Review · Reviewer_BMhR · 2021-07-17

**Rating:** 7
**Confidence:** 2

**Summary:**

In this paper, the authors propose a new method to allow video encoder optimization in training temporal action localization (TAL) models, which is often ignored by existing TAL methods due to GPU memory constraints. The proposed method LoFi reduces the demanding GPU memory requirement by using a lower temporal and/or spatial resolution in the mini-batch construction. The proposed LoFi technique has shown to help enhance the performance of off-the-shelf TAL models on ActivityNet and HACS-v1.1. Also, the performance improvement is universal across different types of video encoder.

**Limitations And Societal Impact:**

Yes.

**Main Review:**

The paper is well-motivated.

Sparse temporal sampling and different spatial scales may have been widely adopted as data augmentation techniques when training action classification models with relatively short videos. However, for temporal action localization tasks on long videos, the frozen video encoder can be a bottleneck to the model performance. LoFi is indeed a simple yet effective method to allow adaptation of the pre-trained video encoder to the target video domain of the downstream TAL task.

Training with long videos is always resource-demanding, for different kinds of video understanding or video+language understanding tasks.  I believe that the proposed method LoFi may have the potential to be extended to other tasks on long videos other than temporal action localization.


**Time Spent Reviewing:**

2

---

> ### Author Response · Authors · 2021-08-10
> **Response to Reviewer BMhR**
>
> We thank the reviewer for the positive and detailed review. As the reviewer said (we share the same view and observation), using a pre-trained action recognition model as a frozen video encoder for TAL has been a widespread practice but very little studies (we are not aware of any) have questioned its effectiveness. Interestingly our study shows that we can obtain meaningful performance gain by improving this component with a simple and effective method. We consider this should be useful to the community. Finally, we really thank the reviewer for appreciating our efforts on this under-studied video encoder problem in TAL.

---

### Official Review · Reviewer_6HSH · 2021-07-19

**Rating:** 6
**Confidence:** 3

**Summary:**

** Update 8/16/21 **

Thanks to the authors for their rebuttal in response to my questions/concerns. After reading the rebuttal and the other reviews, I am updating my score to 6, to better reflect the contributions made by the work.

**

The paper aims to tackle the hardware constraint for end-to-end optimization of both video encoding and TAL head models. Different from prior works that freeze the video encoding network when training the TAL head, the proposed method optimizes both of them jointly on low-resolution (spatially and temporally) video frames and then finetunes the TAL head on full-resolution videos. The main contribution of this paper is resolving the task discrepancy problem for the video encoder when training TAL models. Extensive experiments are conducted in the paper to compare the lofi setting with several state-of-the-arts, as well as similar models under different pretraining conditions. Results indicate the proposed method outperforms the baselines.


**Ethical Concerns:**

None.

**Limitations And Societal Impact:**

The limitations of this work are stated above. The task discrepancy issue is worth investigating and the authors are encouraged to continue their efforts on the research problem.

The authors do a reasonable job to address the societal implications of the work.

**Main Review:**

The authors would like to tackle the task discrepancy problem and meanwhile bypass the GPU memory limitations for joint end-to-end optimization, so they train the model on low-fidelity data. It’s practical, but lacks sufficient novelty in my opinion. The solution proposed in the paper would generally be considered a data-augmentation technique that expands the dataset by sampling snippets/frames from full-resolution data and reduces GPU usage. The downsampling augmentation is common in image classification or localization tasks, but citations for this are not adequate. (ex. A survey on image data augmentation for deep learning.) Also, the C-LoFi method is too similar to [51]. The resampling method and the training schedule are, though used differently for the TAL task for GPU memory issue, the same as the literature.

Maybe it’s not appropriate to regard this work as a pretraining task and compare it with pretraining baselines. In reviewing previous works, perhaps it’s better to view this work from the angle of data augmentation. It’s also interesting to see if lofi works for other tasks.

Technical soundness:

The author presents in-depth experiments and discussions regarding the lofi setting and other related methods. The tradeoff between accuracy and efficiency can also be found in the paper. Indeed, the proposed method does outscore the baselines. However, it’s not clear what the reason is, whether because the task discrepancy problem is mitigated or just because the data is augmented and the model has more learnable parameters. The experiments are not fair enough because the lofi setup is accessible to the augmented data in the target domain.

Solving the task discrepancy problem is crucial in transfer learning, as we hope the knowledge can be transferred to new domains but don’t want the discriminative features to be wiped out. More efforts need to be put into the investigations and experiments on it. Maybe one possible way is to compare the lofi setting with the model trained end-to-end on full-resolution data, compare with a ACP setting but with a comparable number of learnable weights, or visualize the features from the video encoder.

Clarity of exposition:
The exposition is clear and it’s not hard for readers to get to the point. The proposed method can be easily reproduced.
Perhaps the word “low-fidelity” is a bit confusing because readers might associate it with the model, not the data.

Significance:

The results show that the lofi setup outperforms other pretraining procedures. However, the claim that the proposed method mitigates the task discrepancy problem is not well-supported. If lofi setting were comparable with high-fi training, or is shown relatively advantageous in terms of the accuracy-efficiency tradeoff in the experiments, practitioners can benefit and build on the proposed idea.
On the other hand, the results will be more important if not only TAL, but other related tasks are addressed using the same lofi setting.




**Time Spent Reviewing:**

2-4 hours

---

> ### Author Response · Authors · 2021-08-10
> **Response to Reviewer 6HSH**
>
> We thank the reviewer for the detailed review as well as the suggestions for improvement. Our response to the reviewer’s comments is as below:
>
> ---
>
> Q1: _It’s practical, but lacks sufficient novelty._
>
> R1: We have extensive experiments showing that our solution is effective, clearly surpassing more complex and more computationally expensive solutions. In our opinion, that the proposed solution is also intuitive and simple should be seen as a plus, not a minus. Furthermore, a multitude of papers argue that end-to-end training is not possible for TAL. Challenging this conventional and widespread belief is, in our opinion, a meaningful contribution on its own. Finally, we would like to stress again the contributions listed in L67-76 in our main paper.
>
> ---
>
> Q2: _The solution proposed in the paper would generally be considered a data-augmentation technique. [...] The downsampling augmentation is common in image classification or localization tasks, but citations for this are not adequate._
>
> R2: Unfortunately we are not aware of any such work. The only data augmentation technique known to us (we used in this work) is that of jittering the scale, but we consider it as rather unrelated to our approach. Our approach does not add variability to the input through some scale jitter. Instead, we strongly downsize the spatial and/or temporal dimensions, making the train-time and test-time dimensions more dissimilar. Note that our method, as well as all our baselines, use the standard scale jitter augmentation.
>
> ---
>
> Q3: _It’s also interesting to see if lofi works for other tasks…  The results will be more important if not only TAL, but other related tasks are addressed using the same lofi setting._
>
> R3: We believe that our method is not TAL specific but generally applicable to other tasks (e.g., video captioning and video question answering) that suffer from the task discrepancy problem during training (Reviewer BMhR has the same view too). In this study, we focus on TAL so that extensive in-depth analysis can be conducted as we did in both the main paper and supplementary document. We will examine more tasks in the future.
>
> ---
>
> Q4: _It’s not clear what the reason (of improvement) is, whether because the task discrepancy problem is mitigated or just because the data is augmented and the model has more learnable parameters.  The experiments are not fair enough because the lofi setup is accessible to the augmented data in the target domain._
>
> R4: We would like to emphasize that our method does not add any learnable parameters. We will revise the document to make sure no phrase suggests otherwise. Also, we believe our method does not present any advantage from the data augmentation perspective (see R2 above). As previously argued, the spatial and temporal dimensions at training and at test time become dissimilar. The data augmentation techniques used are otherwise the standard ones, including scale jitter. We stress that our experiments are fair as all the methods we compare against use the same training data, including the target domain data.
>
> ---
>
> Q5: _If lofi setting were comparable with high-fi training_
>
> R5: Unfortunately producing a full fidelity model is not straightforward - that is indeed the main limitation our paper is trying to tackle. To be fair, we had trouble training a full fidelity model, which we attribute to a very low batch size (which is a known problem). To solve this issue, we would need to implement training distributed on multiple machines. Thus we believe this might be out of the scope of the rebuttal. We nonetheless think it is an interesting proposal and we will run the experiments in the future for completeness.
>
> ---
>
> Q6: _accuracy-efficiency tradeoff_
>
> R6: Thanks for this great comment. We have now compared the training time of the video encoder by ACP and LoFi in the table below. We use a machine with 4 V100 GPUs.
>
> | Method | Dataset | Backbone | Time/epoch | Epochs | Training time | AmAP |
> | --- | ----------- | ----------- | ----------- | ----------- | ----------- | ----------- |
> | ACP | Kinetics400 | ResNet50 | 2696 sec | 100 | 74.9 hr | 34.26 |
> | LoFi | ActivityNet1.3 | ResNet50 | 2694 sec | 20 | 15.0 hr | 34.96 |
> | LoFi | ActivityNet1.3 | ResNet34 | 1624 sec | 20 | 9.0 hr | 34.74 |
> | LoFi | ActivityNet1.3 | ResNet18 | 1075 sec | 20 | 6.0 hr | 34.49 |
>
> We make these observations: (1) ACP and LoFi take similar per-epoch training time; (2) LoFi is 5 times faster to converge; (3) Using lightweight backbones can significantly reduce the training time at some performance cost. We will add this experiment to the revised version.

---

### Decision · Program_Chairs · 2021-09-28

**Decision:**

Accept (Poster)

**Comment:**

This paper presents work on temporal action localization.  The main idea is to use low fidelity (e.g. lower temporal resolution) to enable end-to-end training within the constraints imposed by large video batches / models in GPU memory.  The reviewers appreciate the simplicity and effectiveness of this idea.  While based on related efforts in training image and other video models, identifying this approach to enable end-to-end training is interesting to the researchers working in this field, challenging a standard assumption about the difficulties in end-to-end training.


**Consistency Experiment:**

NeurIPS has a long history of experimentation. In 2014, NeurIPS ran an experiment in which 10% of submissions were reviewed by two independent committees to quantify the randomness in the review process. This year, we repeated a variant of this experiment to see how the quality of the review process has changed over time.  This paper was part of the experiment and was therefore assigned to two committees (consisting of reviewers, an Area Chair, and a Senior Area Chair) that reached independent decisions.  If both committees made the same recommendation, this recommendation was followed. If a single committee recommended acceptance, the paper was accepted (with the exception of a few cases in which the other committee identified what we considered a fatal flaw, e.g., an error in a key result).

This copy’s committee reached the following decision: **Accept (Poster)**

The other committee assigned to the paper recommended **Reject**.  You can find the other set of reviews, along with any follow up discussion with the authors here:
https://openreview.net/forum?id=iNf3V0m57wz